# High Expression of *PRNP* Predicts Poor Prognosis in Korean Patients with Gastric Cancer

**DOI:** 10.3390/cancers14133173

**Published:** 2022-06-28

**Authors:** Minseok Choi, SeongRyeol Moon, Hyo Jin Eom, Seung Mook Lim, Yon Hui Kim, Seungyoon Nam

**Affiliations:** 1College of Medicine, Gachon University, Incheon 21565, Korea; 201838427@medicine.gachon.ac.kr; 2Department of Health Sciences and Technology, Gachon Advanced Institute for Health Sciences and Technology (GAIHST), Gachon University, Incheon 21999, Korea; moon0620@gachon.ac.kr; 3Department of Genome Medicine and Science, AI Convergence Center for Medical Science, Gachon Institute of Genome Medicine and Science, Gachon University Gil Medical Center, Gachon University College of Medicine, Incheon 21565, Korea; 4Research and Development Department, Corestem Inc., Seongnam 13486, Korea; hyojin3149@gmail.com; 5Department of Biomedical Science, CHA University, Seongnam 13486, Korea; lsmook17@naver.com; 6HieraBio Inc., Seongnam 13605, Korea; sarahkim@hierabio.com

**Keywords:** *PRNP*, gastric cancer, epithelial mesenchymal transition, prion protein, prognosis factor, gene set enrichment analysis

## Abstract

**Simple Summary:**

Gastric cancer is a lethal cancer that is prevalent in East Asia. It is critical to secure prognostic markers for monitoring patients with GC. Recently, *PRNP*, the gene encoding prion protein PrP, has been associated with cell proliferation in diverse cancer types. However, the value of *PRNP* as a prognostic factor for patients with GC has yet to be inspected. The aim of our study was to inspect *PRNP* gene expression in terms of a prognostic value in GC by utilizing publicly available large GC cohorts with information on survival and gene expression profiles. As a result, we found that *PRNP* high- vs. low-expressing patients with GC showed poor survival probability in Korean GC cohorts and that knockdown of *PRNP* decreased cell viability of GC cells. These findings provide evidence for *PRNP* as a valuable tool for follow-up in patients with GC.

**Abstract:**

Gastric cancer (GC) has the highest occurrence and fourth-highest mortality rate of all cancers in Korea. Although survival rates are improving with the development of diagnosis and treatment methods, the five-year survival rate for stage 4 GC in Korea remains <10%. Therefore, it is important to identify candidate prognostic factors for predicting poor prognosis. *PRNP* is a gene encoding the prion protein PrP, which has been noted for its role in the nervous system and is known to be upregulated in various cancers and associated with both cell proliferation and metastasis. However, the value of *PRNP* as a prognostic factor for Korean GC patients remains unclear. Here, we analyzed the relationship between *PRNP* expression and survival in three independent datasets for Korean patients with GC as well as the TCGA-STAD dataset. Survival analysis indicates that high levels of *PRNP* expression are associated with poor overall survival of patients with GC. Gene set enrichment analysis showed that *PRNP* is associated with epithelial mesenchymal transition and Hedgehog signaling. In addition, proliferation of GC cell lines was inhibited after siRNA-mediated knockdown of *PRNP*. In conclusion, our study suggests a potential role for *PRNP* as a candidate prognostic factor for patients with GC.

## 1. Introduction

Gastric cancer (GC) has the highest occurrence and fourth-highest mortality rate in Korea [1] and the five-year survival rate for stage IV GC is <10%, which is unsatisfactory [1]. Efforts to optimize existing chemotherapies and to develop targeted therapies are expected to increase survival rates [2]. Representative targeted therapies for GC include trastuzumab, a monoclonal antibody that targets HER2, and ramucirumab, which targets VEFG-2. Targeted therapies aimed at EGFR, HGFR, and VEGFR are currently under development [3]; however, their therapeutic effects in GC may be different from patient to patient owing to the molecular heterogeneity of GC [4]. Therefore, it is important to identify factors that can predict poor prognosis in order to facilitate individual treatment choice [5].

*PRNP* is a gene encoding the protein PrP, also known as CD230 [6]. PrP is expressed in a range of tissues, especially the nervous system, and is involved in prion disease [7]. It is also involved in various nervous system processes, such as central nervous system development and neuron survival [8]. Recent studies show that PrP is associated with cancer [9]; indeed, upregulated PrP expression has been observed in various cancers, including GC, breast cancer, colorectal cancer, and pancreatic cancer [10,11,12,13,14,15]. PrP overexpression is associated with a poor prognosis, dysregulated cell proliferation, invasion, metastasis, and drug resistance in cancer cells [10,11,12,13,14,15,16].

Few studies have investigated the potential value of PrP as a prognostic factor for GC and the results are inconsistent [17,18,19]. For example, Pan et al. [19] and Liang et al. [18] showed that PrP overexpression could promote tumorigenesis, proliferation, invasion, and metastasis in GC. In contrast, Tang et al. [17] reported that, in GC, negative PrP expression was associated with poor survival rate. Therefore, studies including large GC cohorts as well as clinical association studies between *PRNP* expression and prognosis are necessary.

In this study, we examined the clinical relevance of *PRNP* expression to survival in four publicly available large GC cohorts. We also performed gene set enrichment analysis (GSEA) [20] and proposed potential transcriptional networks for *PRNP* in GC using the transcriptomics of the cohorts.

## 2. Materials and Methods

### 2.1. Collection of mRNA Expression Data and Clinical Information

The mRNA expression data of three Korean cohorts with GC were collected from the Gene Expression Omnibus (GEO, https://www.ncbi.nlm.nih.gov/geo/; accessed on 4 May 2021) database [21], and patient clinical information was collected from the study by Oh et al. [22]. GSE62254 (ACRG cohort) was provided by the Asian Cancer Research Group with GC patient data collected at Samsung Medical Center (Seoul, South Korea) [22,23]. GSE26942 (KSKG cohort) includes GC patient data collected from Kosin University Gospel Hospital (Busan, South Korea) and Korea University Guro Hospital (Seoul, South Korea) [22]. GSE13861 (YUSH cohort) data were collected at Yonsei University Severance Hospital (Seoul, South Korea) [22,24]. For validation of survival analysis and Cox regression analysis, mRNA expression levels and clinical information from The Cancer Genome Atlas Stomach Adenocarcinoma (TCGA-STAD) cohort [25] were collected from cBioPortal (https://www.cbioportal.org/; accessed on 4 May 2021) [26]. For survival analysis, patients with unknown survival periods and statuses were excluded.

### 2.2. Kaplan–Meier Survival Analysis and Cox Proportional Hazards Model

The "Survival" package [27] in R software was used for statistical analyses. For survival analysis, we divided each cohort into “high-*PRNP”* and “low-*PRNP”* groups based on the median *PRNP* gene expression level. We analyzed the survival rates of the two groups using the Kaplan–Meier method and compared the survival curves by using the log-rank test. We used a multivariate Cox proportional hazards model to obtain age- and gender-adjusted hazards ratios. The variable “age” was used after classification into two categories (≥60 and <60) [28]. Results where *p* < 0.05 were considered statistically significant.

### 2.3. Gene Set Enrichment Analysis (GSEA)

We used GSEA (version 4.2.2) [20] software to identify differences in biological function according to *PRNP* expression in Korean patients with GC. The GSEA analysis was performed in two groups divided according to median *PRNP* expression levels in each of the three Korean cohorts (ACRG, KSKG, and YUSH), as in the survival analysis. The hallmark gene set (v7.5.1) [29] of the Molecular Signatures Database (MSigDB) [30] was used as the reference gene set, and default values were used for all parameters. Enrichment analysis was considered significant when the false discovery rate (FDR) was <0.25 [20].

### 2.4. Differential Expression Gene Network

PATHOME-Drug [31] is a simple statistical test for evaluating the significance of differential expression patterns along sub-pathways using the Kyoto Encyclopedia of Genes and Genomes (KEGG) pathway database. We divided the patient groups from the ACRG, KSKG, and YUSH cohort data into “high-*PRNP”* and “low-*PRNP”* groups and evaluated the sub-pathways in which the expression pattern was differentially changed between the two groups. We screened for sub-pathways that overlapped in two or more of the three cohorts. The potential interactions of *PRNP* and *RHOA* with genes in the selected sub-pathways were constructed using the STRING database [32]. The constructed network data were manually curated and visualized using Cytoscape [33].

### 2.5. RNA Extraction and Real-Time qPCR

Total RNA was isolated using TRIzol reagent (Ambion, Texas, USA). RNA quantification was performed using a NanoDrop^TM^ spectrophotometer (Thermo Fisher Scientific, Wilmington, DE, USA) according to the manufacturer’s protocol. All real-time PCRs were performed using SYBR Green Master Mix (Bio-Rad Laboratories Inc., Hercules, CA, USA). Sample amplification was performed using CFX384 (Bio-Rad Laboratories, Inc., Hercules, CA, USA). β-actin was used as a normalization control. Results are expressed as fold changes calculated using the ΔΔCt method for the control samples. The experiment was carried out in triplicate, and the results are expressed as mean values. The primer sequences were as follows: β-actin, 5′-ggacttcgagcaagagatgg-3′ (forward) and 5′-agcactgtgttggcgtacag-3′ (reverse); *PRNP*, 5′-acaactttgtgcacgactgc-3′ (forward) and 5′-tggagaggagaagaggacca-3′ (reverse).

### 2.6. siPRNP Transfection and MTS Cell Viability Assay

The human GC cell lines SNU-216, SNU-620, SNU-668, SNU-601 (KCLB, Seoul, Korea), AGS (ATCC, Mansfield, VA, USA), and MKN-1 (RIKEN, Tsukuba, Japan) were cultured in RPMI 1640 (Invitrogen, Carlsbad, CA, USA) medium supplemented with 10% serum under fasting conditions. The identities of the cell lines were verified by short tandem repeat profiling (ATCC). After seeding 3.0 × 10^3^ cells in a 96-well plate, they were cultured for 24 h. Transfection of siRNA was performed in 20% OPTI-MEM plus 80% culture media for 72 h. In this experiment, DharmaFECT 1 Transfection Reagent (T-2001-04, Dharmacon/Thermo Fisher Scientific, Waltham, MA, USA), ON-TARGETplus Non-targeting Pool (D-001810-10-50, Dharmacon/Thermo Fisher Scientific), and ON-TARGETplus SMARTpool *PRNP* siRNA (L-011101-00-0005, Dharmacon/Thermo Fisher Scientific) reagents were used. Absorbance at 490 nm was recorded 3.5 h after the addition of 40 μL/well of CellTiter 96 AQueous (G3581, Promega Corporation, Madison, WI, USA). A negative control used “scrambled” siRNA. The experiment was repeated thrice.

## 3. Results

### 3.1. Overview

An overview of this study is shown in Figure 1. First, the open cohorts (GSE62254, GSE26942, and GSE13861) provided by GEO were divided into high- and low-*PRNP* groups based on the median *PRNP* expression levels. Kaplan–Meier survival analysis and Cox proportional hazard model analysis were performed for each patient. GSEA [20] was performed for functional analysis and a differential expression network was drawn using PATHOME-Drug [31] and STRING [32]. Finally, we investigated how silencing of *PRNP* affected cell growth in GC cell lines. 

### 3.2. High Levels of PRNP Expression Are an Independent Prognostic Factor for GC

Each of the four cohorts (ACRG, KSKG, YUSH, and TCGA-STAD) was divided into a high-*PRNP* and a low-*PRNP* group by using the cutoff of the median value of *PRNP* expression in 300 ACRG patients, 202 KSKG patients, 65 YUSH patients, and 377 TCGA patients (Appendix A). Log-rank tests revealed significant differences in survival rates between the high-*PRNP* group (red) and the low-*PRNP* group (blue) in all cohorts (Figure 2) where the high-*PRNP* group had lower survival rates than the low-*PRNP* group. We used the Cox proportional hazards model to consider confounding factors, such as age and sex, in the survival analysis, after which the high-*PRNP* group retained lower survival rates than the low-*PRNP* group. In the ACRG cohort, the high-*PRNP* group had a survival rate 1.43 times lower than that of the low-*PRNP* group (95% confidence interval [CI], 1.04–1.98; *p* = 0.0029; Figure 2a). In the KSKG cohort, the high-*PRNP* group had a 1.67 times lower survival rate than the low-*PRNP* group (95% CI, 1.08–2.58, *p* = 0.021; Figure 2b). In the YUSH cohort, the high-*PRNP* group had a survival rate 2.95 times lower than that of the low-*PRNP* group (95% CI, 1.34–6.48, *p* = 0.0071; Figure 2c). In the TCGA cohort, the high-*PRNP* group had a survival rate 1.48 times lower than that of the low-*PRNP* group (95% CI, 1.07–2.05, *p* = 0.019; Figure 2d).

### 3.3. Upregulation of PRNP in GC Is Associated with Epithelial Mesenchymal Transition, Hedgehog Signaling, and Angiogenesis

We performed GSEA to investigate the biological role of *PRNP* in GC (Figure 3). Each of the three Korean GC datasets was divided into two groups according to the level of *PRNP* expression and a MSigDB [30] hallmark gene set (*n* = 50) [29], which summarized a well-defined specific biological state or process [29]. We selected significantly enriched tumor-associated pathways (FDR < 0.25; Appendix A). Consequently, the epithelial–mesenchymal transition (EMT; Figure 3a), Hedgehog signaling (Figure 3b), and angiogenesis (Figure 3c) pathways were significantly enriched in the high-*PRNP* group. EMT, Hedgehog signaling, and angiogenesis pathways are involved in mechanisms underlying cell proliferation and migration in cancer [34,35].

### 3.4. Networks of Altered Sub-Pathway Genes Reveal Potential Interactions between PRNP and RHOA

The PATHOME-Drug [31] tool was used to evaluate differentially varying sub-pathways between the high-*PRNP*-and low-*PRNP* groups (Figure 4a). Sub-pathways of various biological pathways were differentially altered between the high- and low-*PRNP* groups. Interestingly, the sub-pathways of the “JAK-STAT signaling pathway” and “Wnt signaling pathway” related to the cell cycle [36,37], the “regulation of the actin cytoskeleton”, and the “regulation of cell migration” [38] were differentially altered in all three groups (Figure 4a). We used the STRING tool to identify potential interactions between *PRNP* and genes in the altered sub-pathways (Figure 4b and Appendix A). In a previous GC study [39], RHOA was identified as a potential biomarker for Asian GC. Manually curated networks showed that *PRNP* potentially interacts with *RHOA* through interactions with *GSK3B* and *CSNK2A2* (Figure 4b).

### 3.5. Downregulation of PRNP by siRNA Suppresses GC Cell Proliferation

We evaluated cell growth using the MTS assay to determine whether downregulation of *PRNP* expression could inhibit the proliferation of GC cell lines SNU216, SNU601, SNU620, SNU668, AGS, and MKN1 (Figure 5). First, to confirm that *PRNP* gene expression was effectively suppressed after *PRNP*-specific siRNA (si*PRNP*) transfection, real-time qPCR was performed (Figure 5a). When GC cell lines were transfected with si*PRNP*, *PRNP* expression was reduced compared to the control group (scrambled siRNA) from a minimum of 38% (AGS) to a maximum of 98% (MKN1). Next, cell viability was analyzed according to the downregulation of *PRNP* expression. MTS assay results showed that cell survival decreased in five of the six GC cell lines (SNU216, SNU601, SNU620, SNU668, and MKN1; Figure 5b). These results suggest that the overexpression of *PRNP* is associated with cell proliferation in GC.

### 3.6. PRNP Is Upregulated in the Mesenchymal Phenotype

Cancer cells acquire the characteristics of mesenchymal cells with improved motility and invasion through EMT, which is a mechanism of cancer progression [40,41]. Oh et al. [22] used gene expression data to divide GC into mesenchymal phenotype (MP) and epithelial phenotype (EP) subgroups using an unsupervised hierarchical clustering analysis. We compared the expression level of *PRNP* in the phenotypes of GC classified according to the criteria of Oh and colleagues [22]. We confirmed that the expression levels of *PRNP* were significantly higher in the MP subgroup than the EP subgroup (Figure 6a–c).

## 4. Discussion

In our study, high levels of *PRNP* expression in patients with GC were associated with lower rates of survival in all four cohorts (Figure 2), supporting the findings of Tang et al. [17] and Pan et al. [19]. Our GSEA analysis indicates that genes involved in EMT, Hedgehog signaling, and angiogenesis pathways related to cell proliferation and migration were enriched in the high-*PRNP* group (Figure 3). This is consistent with the function of PrP in promoting proliferation, invasion, and metastasis of GC cells reported in previous studies [18,19].

Liang et al. reported that a PrP-induced increase in cyclin D1 expression in GC induces cell cycle promotion [18]. However, in our study, the expression of *CCND1,* which encodes cyclin D1, was significantly decreased in the high-*PRNP* group in two cohorts (ACRG and KSKG) compared to the low-*PRNP* group (*p* < 0.05).

Besnier et al. reported that PrP modulates the “Wnt signaling pathway” during intestinal epithelial cell proliferation [42]. In a pathway analysis using PATHOM-Drug, the sub-pathways of the “Wnt signaling pathway” were significantly altered in all three cohorts. Network analysis confirmed that *PRNP* interacts with the “Wnt signaling pathway” through *CSNK2A2* and *GSK3B*.

This study reveals a potential interaction between *PRNP* and *RHOA* through altered sub-pathway gene networks. Additionally, we recently highlighted the importance of RHOA in relation to the “activation of invasion and metastasis” in GC signaling [43]. Cell invasion associated with cancer progression requires EMT [44], and the activation of RHOA via TGFβ1 signaling induces GC cell migration via EMT [45]. The EMT signaling pathway was significantly enriched in the high-*PRNP* group (Figure 3). In addition, the expression of *PRNP* was upregulated in the MP subgroup compared to the EP subgroup (Figure 6). These results show the potential of *PRNP* as a prognostic factor for cancer progression associated with the mesenchymal phenotype.

GC cell line experiments show that downregulation of *PRNP* expression after si*PRNP* treatment inhibited the proliferation of GC cell lines (Figure 5). This suggests the potential of *PRNP* as a therapeutic target for cancer treatment. Recently, due to the relevance of PrP in cancer growth and metastasis [19], attempts have been made to utilize PrP as a target for cancer treatment [46]. Interaction between PrP and Hsp70/Hsp90-organizing protein (HOP) is associated with lower survival and greater proliferation in glioblastoma [46], and disruption of PrP-HOP binding inhibits the growth of glioblastoma and improved overall survival [46].

Zhou et al. performed immunohistochemistry (IHC) staining for PrP protein expression among 238 patients who underwent GC surgery, demonstrating a poorer prognosis for high PrP expressing patients than low PrP expressing patients (log-rank test, *p*  <  0.001) [47]. In addition, PrP was expressed at higher levels in metastatic GC than in non-metastatic GC [19]. In other cancer types, regarding PrP protein expression in pancreatic ductal adenocarcinoma patients, the PrP-positive group had a poorer prognosis than the PrP-negative group (log-rank test, *p* < 0.0001) [48]. IHC staining for PrP in colorectal cancer (CRC) patients showed that patients with high PrP expression had a poorer prognosis compared to PrP-negative patients (log-rank test, *p* < 0.0001) [49]. In head and neck squamous cell carcinoma, increased PrP expression was detected in lymph node metastasis compared to the primary lesion [50]. In lung cancer, PrP expression was mostly negative for in situ tumors, whereas PrP was expressed by invasive adenocarcinomas [51].

This study has limitations. We confirmed the correlation between the expression of *PRNP* and the prognosis for Korean patients with GC; however, Guo et al. [52] have previously explained that the correlation between mRNA and protein expression levels is imperfect. Therefore, it is necessary to investigate the correlation between PrP protein expression and clinical outcomes (i.e., survival) by constructing a large retrospective cohort with survival information and using IHC and tissue microarray.

## 5. Conclusions

This study suggests that high-*PRNP* expression is an independent prognostic marker for GC and is associated with cell proliferation and migration. *PRNP* knockdown in GC cell lines inhibited cell viability, but further validation is required to demonstrate the biological function of *PRNP* in GC.

## Figures and Tables

**Figure 1 cancers-14-03173-f001:**
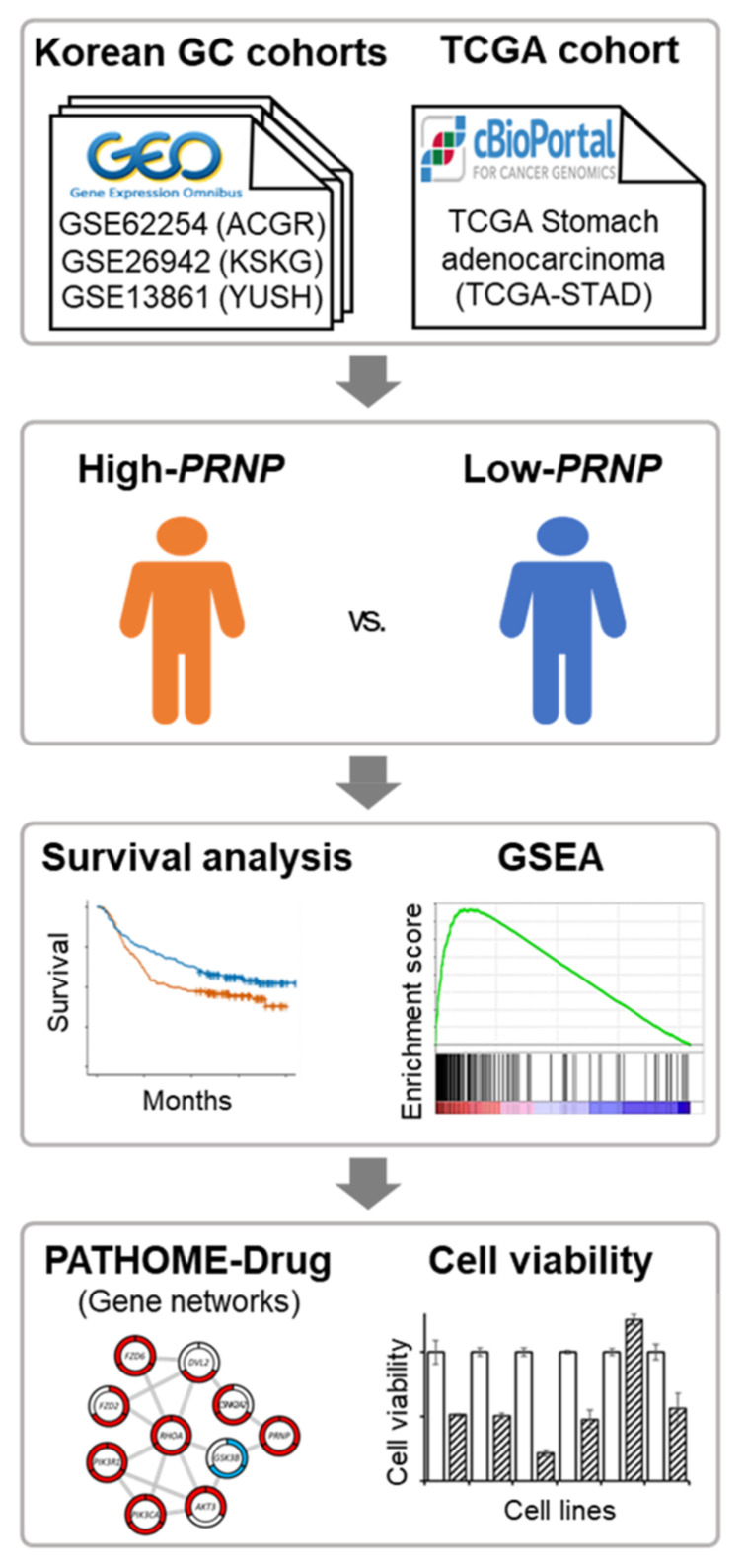
Overview of the study. Three cohorts of Korean GC patients were collected from GEO and divided into “high-*PRNP*” and “low-*PRNP*” groups. To confirm the association between the expression level of *PRNP* and GC prognosis, survival analysis, GSEA, network analysis, and knockdown experiments were performed.

**Figure 2 cancers-14-03173-f002:**
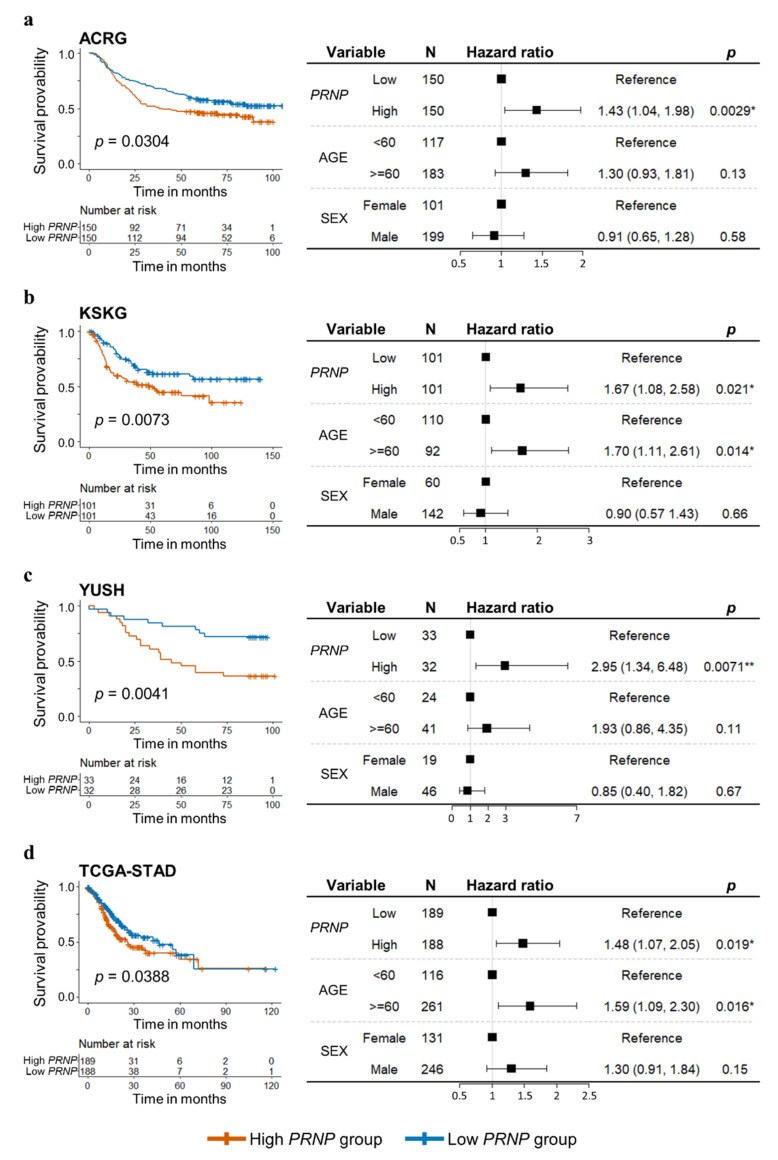
Kaplan–Meier curves, log-rank tests, and Cox proportional hazards model according to mRNA expression of *PRNP*. In all cohorts, the high-*PRNP* group had lower survival rates than the low-*PRNP* group. (**a**) Correlation of *PRNP* expression with overall survival in the ACRG cohort. (**b**) Correlation of *PRNP* expression with overall survival in the KSKG cohort. (**c**) Correlation of *PRNP* expression with overall survival in the YUSH cohort. (**d**) Correlation of *PRNP* expression with overall survival in the TCGA cohort. Black squares represent the hazard ratio. *, *p* < 0.05; **, *p* < 0.01.

**Figure 3 cancers-14-03173-f003:**
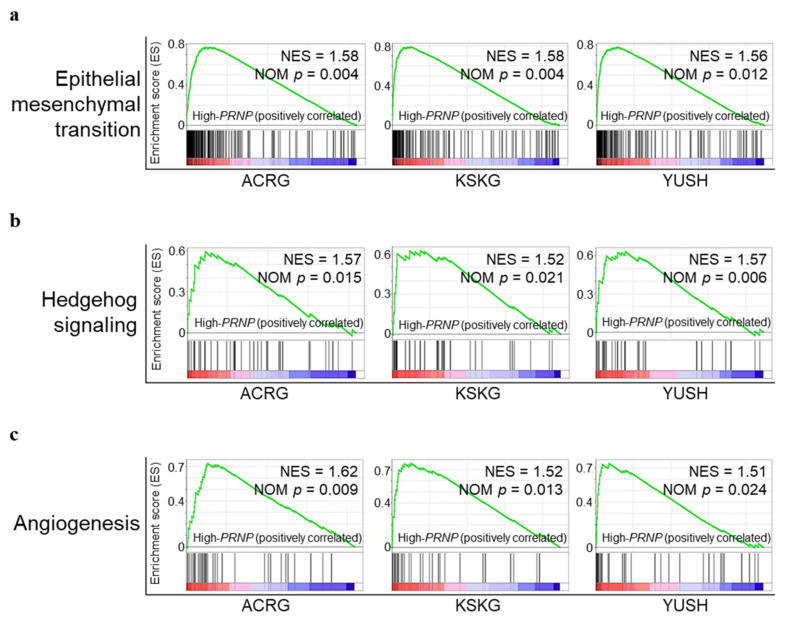
GSEA comparing high-*PRNP* and low-*PRNP* expression groups in three Korean GC cohorts (ACRG, KSKG, and YUSH). Enrichment plot shows important pathways identified using GSEA. (**a**) GSEA indicated the enrichment of EMT-related genes in Korean GC with high *PRNP* expression. (**b**) GSEA showed the enrichment of Hedgehog signaling-related genes in Korean GC with high *PRNP* expression. (**c**) GSEA indicated the enrichment of angiogenesis-related genes in Korean GC with high *PRNP* expression. NES, normalized enrichment score; NOM *p*, normalized *p* value.

**Figure 4 cancers-14-03173-f004:**
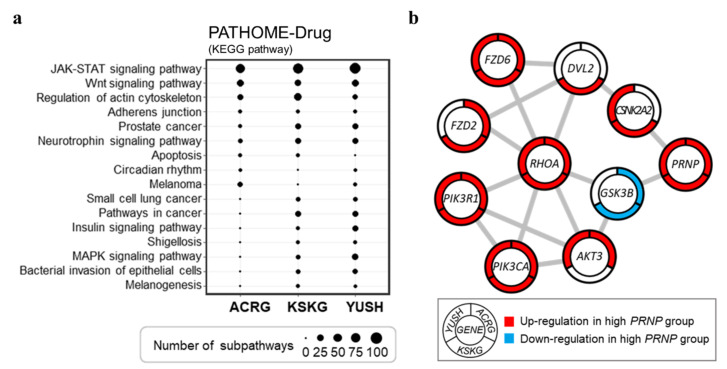
Altered sub-pathway gene networks. (**a**) PATHOME-Drug analysis revealed KEGG pathways with significantly altered sub-pathways in the high-*PRNP* group. The size of each dot represents the number of altered sub-pathways. (**b**) Interactions of significantly altered sub-pathway genes and *PRNP*. A gene name can be found in the center of each node, and the three surrounding regions represent ACRG, KSKG, and YUSH cohorts. Node color indicates a significant difference in the gene expression levels of the low- and high-*PRNP* groups in each cohort (*p* < 0.05). Red indicates that gene expression was significantly upregulated in the high-*PRNP* group compared to the low-*PRNP* group. Blue indicates that gene expression was significantly downregulated in the high-*PRNP* group compared to the low-*PRNP* group. Grey lines represent interactions between each gene (node).

**Figure 5 cancers-14-03173-f005:**
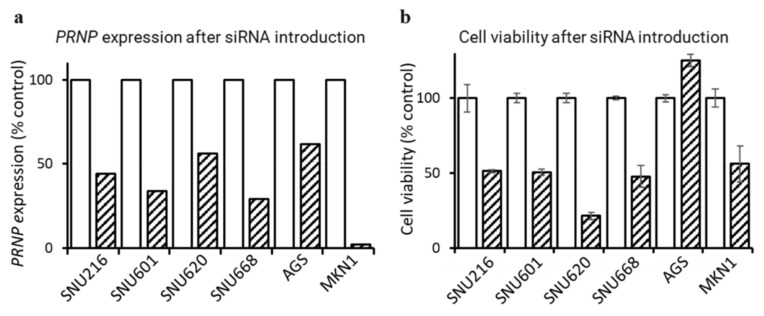
Comparison of gene expression and cell proliferation after si*PRNP* transfection in GC cell lines. (**a**) The expression level of the *PRNP* gene was determined by real-time qPCR in GC cell lines normalized using β-actin as a control and expressed as the average of two replicates. (**b**) Cell viability was determined by MTS assay after si*PRNP* transfection. Data represent mean ± SD of triplicate tests.

**Figure 6 cancers-14-03173-f006:**
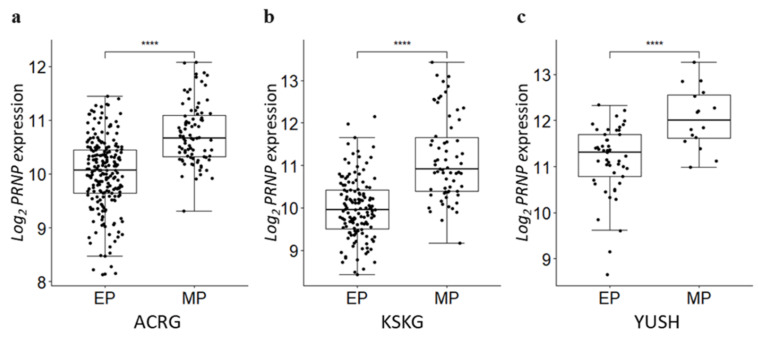
*PRNP* expression according to GC phenotype. (**a**) In the ACRG cohort, MP had significantly upregulated *PRNP* expression compared to EP. (**b**) In the KSKG cohort, MP had significantly upregulated *PRNP* expression compared to EP. (**c**) In the YUSH cohort, MP had significantly upregulated *PRNP* expression compared to EP. •, a patient; ****, *p* < 0.001.

## Data Availability

Publicly available datasets were used for analysis in this study. Data sources are described in Materials and Methods.

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
