# Peer review of "High Expression of PRNP Predicts Poor Prognosis in Korean Patients with Gastric Cancer"

_cancers, 2022, doi:10.3390/cancers14133173_

Round 1

Reviewer 1 Report

Cancers-1765796

The manuscript by Choi et al. reports that PRNP is a potential prognostic biomarker in patients with gastric cacer which is one of the lethal cancers in East Asia. Their conclusions were obtained from Korean gastric cancer cohorts. The manuscript is well-written and the findings described are interesting. However, as described, the authors should investigate the correlation between PrP protein expression and clinical outcomes. I think PrP immunohistochemistry is easy method for this. I believe the authors add immunohistochemical data and strengthen the findings reported.

Reviewer 2 Report

In their original research article, Choi & Moon et al. explore the use of PRNP mRNA levels as a predictor for gastric cancer outcomes. To achieve this, they obtained mRNA expression data from three GEO Korean cohorts along with the corresponding clinical data.

The study is structurally sound, first by investigating PRNP expression in available cohorts, performing survival analysis in groups based on PRNP expression, validating the results in silico by data from TCGA, then performing GSEA to identify ties to other molecules and they finished their study by performing functional experiments using siRNA assays on GC cell lines.

I have no other comments really. Language is good, I couldn't find something to comment, and the text is easy to read. I think this is a good paper that is worthy of being accepted for publication.

Round 2

Reviewer 1 Report

The revised manuscript has greatly been improved.